# Morphological and Functional Remodeling of Vascular Endothelium in Cardiovascular Diseases

**DOI:** 10.3390/ijms24031998

**Published:** 2023-01-19

**Authors:** Ghassan Bkaily, Danielle Jacques

**Affiliations:** Department of Immunology and Cell Biology, Faculty of Medicine and Health Sciences, Université de Sherbrooke, Sherbrooke, QC J1H 5N4, Canada

**Keywords:** endothelium, endothelium physiology, endothelium pathology, endothelium dysfunction, hypertension, endothelium remodeling, endothelium released factors, atherosclerosis, adhesion molecules, calcium, ion transporters

## Abstract

The vascular endothelium plays a vital role during embryogenesis and aging and is a cell monolayer that lines the blood vessels. The immune system recognizes the endothelium as its own. Therefore, an abnormality of the endothelium exposes the tissues to the immune system and provokes inflammation and vascular diseases such as atherosclerosis. Its secretory role allows it to release vasoconstrictors and vasorelaxants as well as cardio-modulatory factors that maintain the proper functioning of the circulatory system. The sealing of the monolayer provided by adhesion molecules plays an important role in cardiovascular physiology and pathology.

## 1. Introduction

The vascular endothelium is much more than just a physical barrier at the interface between the circulating blood and the vascular wall [1,2,3]. In the 1960s, the vascular endothelium was considered a passive barrier protecting the vascular wall from circulating blood [2,3,4]. In the 1980s, it emerged as a multifunctional endocrine organ, playing an essential role in regulating cardiovascular tone [2,5,6,7,8,9]. 

## 2. The Vascular Endothelium

The vascular endothelium is a simple tissue in its morphology but complex in its function. Although it is formed as a single monolayer, it is capable of sensing hemodynamic and rheologic changes as well as responding to these modifications of its environment. Vascular tone is maintained by balancing vasodilating and vasoconstricting factors released by the endothelium. In addition, the endothelium plays a key role in controlling the migration and proliferation of VSMCs [2]. 

The integrity of the endothelial monolayer is essential to regulate vascular permeability and protect the vessel against platelet deposition and thrombus formation. Furthermore, the integrity of this monolayer requires that the morphology and the contacts between ECs do not change. 

## 3. Origin and Differentiation of the Vascular Endothelium

The endothelium is the first cell type to constitute blood vessels. The formation of blood vessels and vasculogenesis result from the differentiation of the mesodermal cells into angioblasts due to the presence of specific proteins. These angioblasts are the precursors of endothelial and blood cells. The physiological primitive angiogenesis takes place to form the vascular tree, which gives birth to buds of the branches that give birth to the heart, including its endocardial endothelial cells. Then, remodeling of the vascular tree takes place to form capillaries and veins, including large arteries [2,10,11]. The endothelium of these newly formed blood vessels differs depending on the type of vessels. For example, fibroblast growth factor (FGF) receptors seem to be expressed only in large vessels [12]. Several ligands and their corresponding receptors are implicated in the differentiation and formation of the endothelium, including vascular endothelial growth factor (VEGF) and its receptors 1 and 2 [10]. 

The heart’s formation is more than a deformation of blood vessels, and differentiation of vascular endothelium into endocardial endothelium forms the left (arterial) and right (venous) endocardial endothelium layers. Both endocardial and vascular endothelium are separated from their muscle cells by a basal lamina membrane [2]. Vasculogenesis occurs during early embryonic development, whereas angiogenesis happens during adulthood. Angiogenesis is usually associated with diseases [11]. Several endothelial markers exist, such as VE-cadherin, PECAM-1, Tie-1 and 2, and flk1. Notch family activation plays an essential role in defining the characteristics and identities of arterial endothelial cells [13]. Although the molecular aspect of the arterial specification is more precise, little is known concerning the venous specification [13]. For example, the vascular endothelium can adapt its function depending on the environment. Still, it does not change its phenotype, such as in transplanted arterial and venous vessel grafts, where the graft vessel’s endothelium matches the host vessels’ characteristics [14,15]. 

## 4. Role of the Endothelium in Vascular Physiology

All blood vessels contain endothelial cells that form the intima. Two types of blood vessels only have endothelial cells: capillaries and venules. The intima is formed by continuous and discontinuous (fenestrated) layers of endothelial cells. Examples of continuous endothelium are arteries and veins. Tight, adherent junctions connect the continuous layer of endothelium side by side. Transport molecules go through this sealed monolayer of the endothelium via a transcytosis mechanism, such as caveolae (caveolin-1) and vesiculo-vacuolar organelles [16]. A fenestrated, discontinuous layer of endothelium permits extensive transport of molecules toward tissues such as the liver.

Several physiological functions are attributed to vascular endothelium, independent of their localization in the vascular tree, such as tuning the level of vascular endothelial cells (VECs) vasoconstrictors and vasorelaxers [2,17], regulation of coagulation and inflammation [3,18], and playing an essential role as a gatekeeper of fatty acids transport [19,20,21,22,23,24].

Among the vasoconstrictive factors released by the vascular endothelium are endothelin-1 (ET-1), thromboxane A2 (TxA2), as well as angiotensin-II (AngII) [2,16,17,25,26]. On the other hand, the most vasorelaxant factors released by the endothelium are nitric oxide (NO), prostacyclin (PGI2), and endothelium-derived hyperpolarizing factor (EDHF) [2,27]. 

The blood vascular system consists of a circuit of vessels in which the continuous movement of the heart pump maintains the blood flow. Blood vessels distribute nutrients, oxygen, and hormones to all organs and tissues and transport the products of cellular metabolism. The walls of arteries and veins, such as the thoracic aorta, used commonly in the literature, consist of three concentric tunics that are firmly joined from the inside out [2] (Figure 1): (1) The intima is the thin innermost layer that lines the various vascular walls, including those of the capillaries and venules. It is composed of a monolayer of endothelial cells (ECs) in direct contact with the blood and forming the vascular endothelium. The ECs provide a smooth inner surface that minimizes friction, which facilitates blood flow. The vascular endothelium is supported by a basal lamina and a thin connective tissue formed by collagen and some elastic fibers; 

(2) The media is the thickest intermediate layer of the vascular wall. It consists of vascular smooth muscle cells (VSMCs), collagen, and elastin. This layer is absent in the capillaries and venules; (3) The adventitia is the outermost layer of the vascular wall. It is absent in capillaries and venules. This layer is formed of supporting connective tissue consisting mainly of collagen. It is also crossed by numerous nerve endings controlling the activity of the muscle fibers as well as the blood vessels feeding the vascular wall, called vasa vasorum (vessels of the vessels). The relative importance of these three layers varies according to the type of blood vessel [2]. In conclusion, all blood vessels have an endothelium but not necessarily adventitia or VSMCs, hence the importance of studying and learning more about the vascular endothelium.

## 5. Structure of the Vascular Endothelium of Arteries and Veins

A monolayer of flat cells forms the vascular endothelium of arteries and veins, with a central nucleus measuring 10–20 µm in diameter. VECs are characterized by extensive intercellular overlap and long, deep slits that contribute to the integrity of the vascular endothelium [2]. The integrity of this monolayer is ensured by a dynamic cytoskeleton [2,28,29,30] as well as by contacts between cells and between these cells and the extracellular matrix [2,31,32]. In vivo and in situ morphology studies have shown the presence of tight junctions, adhesion junctions, and gap junctions between adjacent VECs (including aortic VECs) [32,33,34]. In addition, several roles have been attributed to junctional communication at the vascular endothelium level, including intercellular nutrient exchange, regulation of growth and differentiation, coordination of cellular response to exogenous and endogenous stimuli, and maintenance of vascular tissue homeostasis [35,36,37,38]. 

The cytoskeleton is well-developed in ECs. It contributes to vascular homeostasis and seems to play an essential role in the repair and integrity of these cells [28,29,30]. VECs contain the actin protein in its filamentous polymeric form, called F-actin, and in its globular monomeric form, called G-actin [39,40]. Therefore, F- and G-actin play a role in the shape of ECs. The balance between the monomeric and polymeric forms could be altered during stimulation of ECs and contribute to the modulation of intercellular junctions that affect the vascular permeability of the endothelial layer. Indeed, during the migration of VECs, G-actins increase compared to F-actins [41]. The migration of these cells also involves the redistribution of centrosomes [28]. Actin microfilaments are localized within the cell as short, thin stress fibers and form a continuous band at the periphery [28,39]. In situ studies have also demonstrated the presence of the protein myosin at the level of these microfilaments [42,43], which plays an essential role in cell adhesion, and facilitates the adaptation of the vascular wall to variations in blood flow pressure [28]. The presence or absence of an actin isoform allows the identification of ECs. Therefore, the presence of α-actin in VECs is considered to be a marker for this type of cell [44].

## 6. Role of the Endothelium in Vascular Activity

ECs respond to chemical and physical stimuli by synthesizing and releasing various vasoactive and growth factors [2] (Figure 2). The endothelium possesses anti-adhesive substances that prevent blood from clotting. The anticoagulant and antithrombotic properties of the vascular endothelium, which are essential for vascular homeostasis, are due to the synthesis of vasodilatory factors such as nitric oxide (NO) and prostacyclin [5,16,42,45,46] (Figure 2). On the other hand, the vascular endothelium secretes several vasoconstrictor substances (Figure 2), including endothelin-1 (ET-1), prostaglandins, and several components of the renin-angiotensin system (RAS), such as angiotensin II (Ang II). Ang II [25,26,47,48] and ET-1 [49,50,51] act at the plasma and nuclear membranes of ECs and induce an increase in the intracellular calcium level via activation of their respective receptors, AT_1_/AT_2_ and ET_A_ /ET_B_ receptors. This increase in [Ca^2+^]_i_ may, in turn, modulate the secretory function of ECs [2,52] and survival [26,50]. Furthermore, a balance between the different factors secreted by the EV is essential for maintaining intracellular homeostasis and wall integrity. Any disturbance in this balance leads to endothelial dysfunction, characterized by a decreased capacity for relaxation of the vessel, an increase in the adhesion of blood cells to the vascular wall, and a disturbance in the tunica medial [1,5,9,20,53,54,55,56]. This endothelial dysfunction is generally observed during aging and in several vascular pathologies, such as hypertension, hypotension, atherosclerosis, and heart failure [26,57,58]. All VECs synthesize and secrete von Willebrand factor (vWF), a multifunctional protein involved in the typical arrest of hemorrhage [59]. Indeed, through its interaction with extracellular matrix proteins and membrane receptors, vWF plays a prominent role in blood coagulation, platelet aggregation, and platelet adhesion to the extracellular matrix [60,61]. vWF can also bind to the pro-coagulant co-enzyme, factor VIII, contributing to its stability and, indirectly, to the production of fibrin [60,61]. vWF is stored in small vesicles characteristic of endothelial cells, the Weibel–Palade bodies [60,61,62]. The latter contain other proteins, such as ET-1 [62,63] and interleukin-8 [64]. In addition, vWF is used as a marker of ECs in vitro [65]. 

## 7. Inter-Endothelial Junctions

Junctions between ECs are formed by a transmembrane protein called occludin, which connects to a group of intracellular proteins such as zonula occludin-1 (ZO-1), ZO-2, cingulin, and a new protein linked to GTP, rab13 [66,67]. Their main biological functions are: (1) to form a barrier to paracellular permeability, (2) to maintain the apical-basal polarity of cells, and (3) to assist in intercellular adhesion [68]. In vascular tissue, tight junctions are distributed according to the permeability of the endothelial monolayer in different vascular beds [69]. Figure 2 shows three different types of endothelial junctions. 

The endothelium of the cerebral microcirculation is well sealed (continuous endothelium) with an extensive network of tight junctions between cells to form the blood-brain barrier [70]. ECs are more widely spaced, with relatively fewer tight intercellular junctions in muscle capillaries (fenestrated endothelium) [71,72]. On the other hand, the endothelium in postcapillary venules is highly discontinuous, with very few tight intercellular junctions, to allow more interactions between blood and interstitial tissue (discontinuous endothelium) [69]. 

Adhesion junctions allow ECs to act as strong structural units by linking the cytoskeletal elements of one cell to another [73,74]. These junctions are composed of two categories of proteins: (1) intracellular attachment proteins that connect the junctional complex to the actin filaments and (2) cadherins bind one or more intracellular attachment proteins, and their extracellular domains containing Ca^2+^ binding motifs interact with the extracellular domains of cadherins from another cell. Cadherins allow homophilic and Ca^2+^-dependent adhesion between cells [73,74]. 

The adhesion molecule PECAM is essential for EC activities, such as angiogenesis and control of leukocyte extravasation [75]. Integrins are receptor proteins of importance because they play an essential role in the interaction of ECs with leukocytes during inflammation [76]. The gap junction of ECs consists of connexons in the contact plasma membranes of two cells and forms a microchannel that allows intracellular ions and molecules of low molecular weight (<1000 Daltons) to be exchanged between ECs. One crucial role of gap junctions between ECs is that this microchannel permits the propagation of Ca2+ waves between cells and allows electrical coupling between ECs [77,78]. Connexin types include Cx43, Cx37, and Cx40 [79,80,81,82]. In addition, specific adhesion molecules such as N-cadherin and E-cadherin are essential in establishing coupling between contacting ECs. 

## 8. Ionic Transporters in Vascular Endothelium

In addition to acting as a barrier between circulating blood and vascular smooth muscle cells, the main role of VECs is the secretion of vasoconstrictors and relaxing factors. In turn, these released endothelial vasoactive factors regulate endothelial excitation-secretion coupling. Secretion generally depends on the increase of intracellular Ca^2+^ (Figure 2) [2,26,48]. Calcium plays a significant role in all cell types, including VECs. Regulation of Ca^2+^ homeostasis in vascular endothelium relies on Ca^2+^ influx through Ca^2+^ channels, the ER, mitochondria, and the nucleus, as well as indirectly through the Na^+^/Ca^2+^ exchanger (NCX) [2,27]. In addition, the level of Ca^2+^ release via the ER IP_3_ and ryanodine-sensitive pools highly contributes to the excitation-secretion coupling of VECs [2,27,44]. The level of intracellular Ca^2+^ also depends on the density of the plasma membrane and ER Ca^2+^ pumps [9,83]. 

In the early 1980s, it became apparent that different calcium currents coexisted in several excitable cell types [27,52,83,84]. In VSMCs, two types of calcium currents are present: L-type (high threshold) and T-type (low threshold) [27]. Compared with type L, the T current activates and inactivates relatively quickly from the more negative membrane potentials. Furthermore, the inactivation of the L-type calcium channel is calcium-dependent, whereas the T-type channel is not [27]. There do not appear to be any L- or T-type calcium channels in endothelial cells [27,44,52,85]. In contrast, a voltage-dependent and G-protein coupled calcium channel called the R-type calcium channel has been identified in human VECs, rabbit and human aortic smooth muscle cells, the renal artery, and cardiac ventricular cells [27,44,86,87]. The existence of this calcium channel was initially suggested by Baker et al. (1971) [88]. They showed that this type of calcium channel allows the passive entry of Ca^2+^ into the cell during a long-term depolarization of the membrane. It was named the resting potential calcium channel by DiPolo in 1979 [89]. Unlike the L-type calcium channel, the resting-type (R) calcium channel has no inactivation gates [44,90]. The latter is activated during sustained depolarization of the cell membrane of VSMCs and has a conductance of 24 pS [27,44]. By measuring free intracellular calcium levels using techniques such as microfluorometry and imaging, sustained membrane depolarization produces a rapid, transient increase in intracellular calcium that is followed by a sustained phase [91]. The first phase is abolished by L-type calcium channel blockers such as nifedipine and L- and R-type blockers such as isradipine [87,91]. However, the sustained phase is insensitive to nifedipine, caffeine, and other blockers but sensitive to isradipine and the calcium chelator EGTA [91]. In addition, the R-type calcium channel is responsible for sustained increases in calcium during sustained contraction of vascular smooth muscle in response to various vasoactive and proinflammatory agents such as PAF (Platelet Activating Factor), endothelin-1, as well as insulin [44,90,91]. In addition, calcium influx via the R (resting) channel is primarily responsible for maintaining normal physiological intracellular calcium concentration and the basic tone of vascular smooth muscle [27]. Finally, this channel seems to be involved in basal secretion phenomena of the vascular endothelium [44].

Several calcium channels were suggested to be present in vascular endothelium, such as transient receptor potential channels 1–7 (TRPC1-7), store-operated channels, receptor-operated channels (SOCs), receptor-operated calcium channels (ROC), ryanodine calcium release receptors (RyR), IP_3_ receptor calcium release channels (IP_3_R), and storage-operated calcium entry (SOCE) [89,92,93,94,95,96]. Except for RYR and IP_3_R, all other reported calcium transporters were indirectly suggested to be present in vascular endothelium because it is difficult to prove their presence and contribution to Ca^2+^ homeostasis due to the absence of specific pharmacological blockers and because they cannot be recorded using biophysical techniques. Thus, their presence and contribution to intracellular Ca^2+^ homeostasis are questionable. 

## 9. Morphological and Functional Remodeling of Vascular Endothelium

As mentioned in the previous section, the vascular endothelium secretes substances that cause relaxation and others that cause contraction [6,9,18,21,24,25,97,98] (Figure 2), and a balance between these substances is a significant determinant of vascular homeostasis. Therefore, any activation or damage to endothelial cells, as in cardiovascular diseases such as atherosclerosis, hypertension, and cardiac failure, may cause disequilibrium in their secretory functions and hence contribute to the different symptoms associated with these diseases [6,9,19,20,23,30,92,94,98]. 

In the last decade, several studies investigating endothelial function have provided strong evidence for the role of endothelial dysfunction in both large conduit and small resistance vessels in patients with heart failure [99]. These studies have shown attenuated endothelium-dependent vasodilation in patients with chronic heart failure [99]. Furthermore, a study has shown coronary endothelial dysfunction in patients with new-onset idiopathic dilated cardiomyopathy, suggesting that changes in endothelial function occur early in the course of the disease [100].

Recent work showed that high sodium salt-induced hypertension induced glycocalyx destruction and morphological remodeling in human VSMCs [101]. Since the glycocalyx plays an essential role in VE function, its collapse will cause morphological and functional remodeling, thus affecting VSM function and promoting the remodeling of VSMC [102]. In addition, it was reported in hereditary cardiomyopathy that morphological remodeling takes place early in life before the development of cardiac hypertrophy [103] (Figure 3). Thus, remodeling of the endothelium could be considered a marker of hereditary cardiomyopathy. 

As mentioned, the VECs are separated from the VSMCs by a basal membrane and an internal elastic lamina. However, VECs develop a finger-like projection into the basal membrane, and hypertrophic cell remodeling will promote physical contact between VECs and VSMCs, which promotes cytosolic and nuclear calcium increase. Figure 4 and Figure 5 show examples [104,105].

## 10. Atherosclerosis

Atherosclerosis is a disease that starts at the endothelial cell level. Several major risk factors for atherosclerosis, such as hypercholesterolemia [19,22,24], angioplasty restenosis [56], hypertension, and diabetes [9,20,24], are characterized by abnormal arterial vascular endothelium function, leading to an increase in the inter endothelial cell junction. VECs injury, platelet adhesion/degranulation, invasion of the subendothelium by leucocytes from the circulating blood, and proliferation of contractile and non-contractile VSMCs from the media are critical early events in atherogenesis [9,20,30]. The leading risk factor for atherosclerosis is chronic hypercholesterolemia-induced monocyte attachment to the vascular endothelium, leading to an increase in vascular permeability, which permits entry of immune cells and blood-circulating factors and promotes VSMC proliferation (Figure 6). When immune-inflammatory cells come into contact with VSMCs, they become activated and release cytokines, platelet-activating factor (PAF), tumor necrosis factor-alpha (TNF-α), IL-1, IL6, and eicosanoids. These factors stimulate the release of endothelium factors such as endothelin-1, PAF, and PDGF. In addition, all these factors promote the proliferation of contractile VSMCs (Figure 6). All these factors can be considered pathological first messengers for atherosclerosis. In addition, most, if not all, of these factors induced an increase of intracellular Ca^2+^ via stimulation of R-type calcium channels, leading to secretion and activation of Ca^2+^-dependent intracellular signaling and hypertrophy of VECs.

One of the critical phenomena of atherosclerosis is endothelial denudation. Studies have suggested a loss of gap junctions between ECs in the early development of this disease [105]. Later, gap junctions increase in volume concomitantly with endothelial regeneration [106]. Thus, ECs appear to maintain their gap junctional contact under stressful conditions, even during their migration and proliferation to fill and seal the lesion [107,108]. 

## 11. Conclusions

Knowing the physiology of the endothelium helps us better understand the role of these secretory cells in cardiovascular pathology and to determine their involvement in many diseases, such as hypertension, diabetes, obesity, inflammation, atherosclerosis, and even cancer. This allows us to develop more targeted treatments for these diseases involving the endothelial system. In addition, knowing the different contact systems and their roles in the cardiovascular system’s pathophysiology deserves more studies to design treatments to preserve the endothelial layer seal.

## Figures and Tables

**Figure 1 ijms-24-01998-f001:**
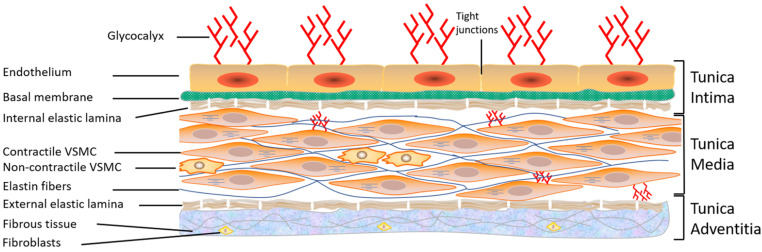
Structure of the vascular wall. Schematic representation showing the three layers of the vascular wall: tunica intima, tunica media, and tunica adventitia, as well as the components of each layer. VSMC: vascular smooth muscle cells (from Bkaily et al., 2021 [2]).

**Figure 2 ijms-24-01998-f002:**
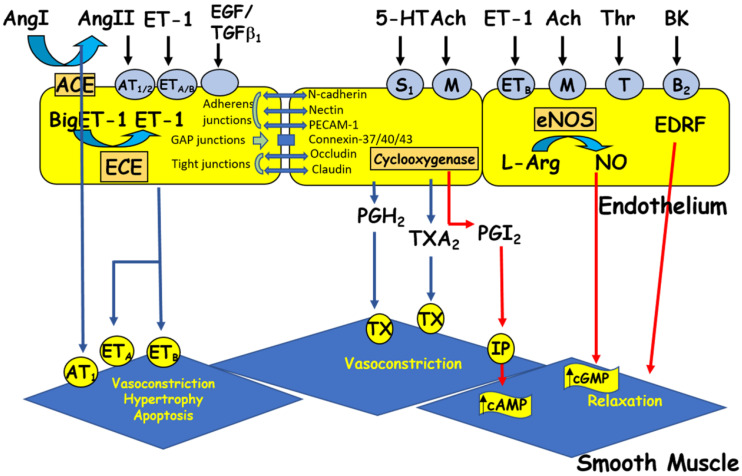
The endothelium produces vasoactive factors that cause either relaxation or contraction of the vascular smooth muscle. Ang I and II: angiotensin I and II, ACE: angiotensin-converting enzyme, Ach: acetylcholine, BK: bradykinin, cAMP/cGMP: cyclic adenosine/guanosine monophosphate, ECE: endothelin-converting enzyme, EDRF: endothelium-derived relaxing factor, ET-1: endothelin-1, 5HT: 5-hydroxytryptamine (serotonin), L-Arg: L-arginine, NO: nitric oxide, NOS: nitric oxide synthase, PGH2: prostaglandin H2, PGI2: prostacyclin, TGFβ1: transforming growth factor beta 1, Thr: thrombin, and TXA2: thromboxane A2. Circles represent receptors (AT: angiotensin receptor, B: bradykinin receptor, ET: endothelin receptor, M: muscarinic receptor, IP: purinergic receptor, S: serotonin receptor, T: thrombin receptor, and TX: thromboxane receptor).

**Figure 3 ijms-24-01998-f003:**
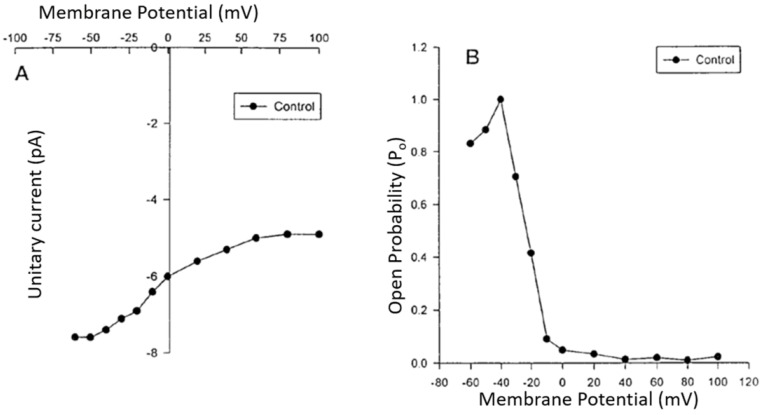
Current to voltage (I/V) relationship curve (**A**) and open probability/voltage relationship (**B**) of the voltage-dependent steady-state R-type Ca^2+^ channel in human aortic VSMCs recorded using the patch clamp technique (modified from Bkaily et al. 1997) [44].

**Figure 4 ijms-24-01998-f004:**
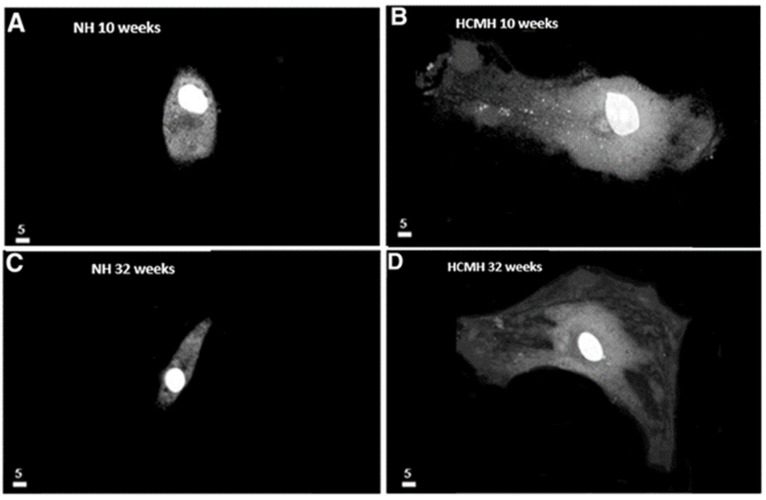
Increase in the volume of endocardial endothelial cells (EECs) in young and old hereditary cardiomyopathic hamsters (HCMHs). (**A**–**D**) Freshly isolated and cultured EECs from 10-week-old normal hamsters (NH) (**A**), 10-week-old HCMH (**B**), 32-week-old NH (**C**), 32-week-old HCMH, and (**D**) show an increase in the volume (in µm^3^) of EECs in HCMHs compared to those of age-matched NHs. In panels A–D, the white scale bar is in µm (modified from Jacques and Bkaily 2019) [104].

**Figure 5 ijms-24-01998-f005:**
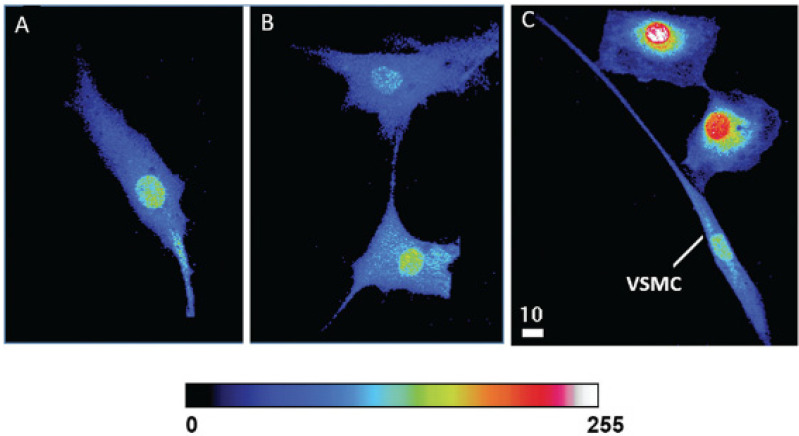
Modulation of cytosolic and nuclear calcium levels of human VSMCs by physical contact with human VECs. Quantitative 3D confocal microscopy images showing the distribution of cytosolic and nuclear Ca^2+^ in hVSMCs in pure culture of hVSMCs without contact (**A**) and in contact with other hVSMCs (**B**), as well as in co-culture with hVECs (**C**). The pseudo-color scale represents the level of fluorescence intensity of the Fluo-3–Ca^2+^ complex from 0 (black, absence of fluorescence) to 255 (white, maximum fluorescence) in panels A, B, and C. The white scale bar value is in micrometers (modified from Hassan et al., 2018) [105].

**Figure 6 ijms-24-01998-f006:**
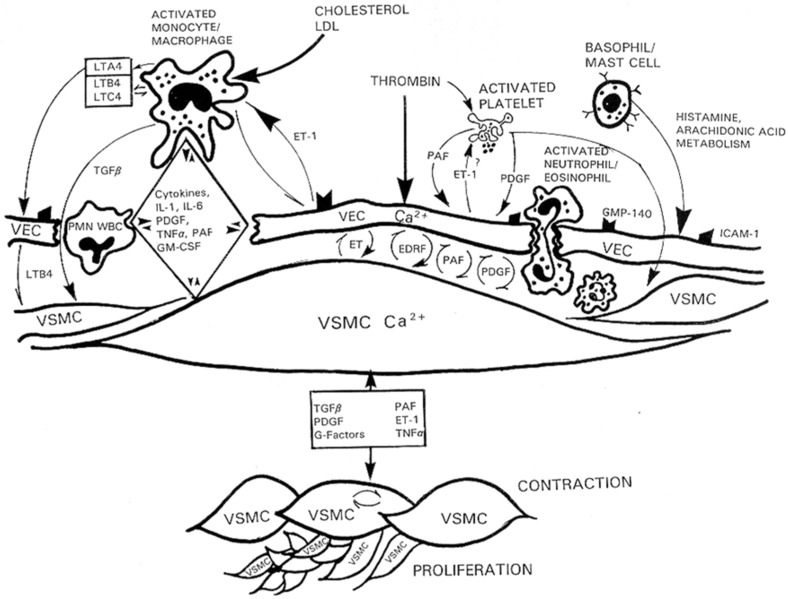
Activation of immune cells induces the release of factors that modulate the function and junctions between VECs, which permits the infiltration of activated neutrophils/eosinophils, and polymorphonuclear cells (PMN). The remodeling of the vascular endothelium induces the secretion of factors that induce either relaxation or contraction of VSMCs. The activated monocytes/macrophages and VECs’ released factors promote the proliferation of contractile and non-contractile VSMCs that characterize atherosclerosis (modified from Bkaily, 1994) [27].

## Data Availability

Not applicable.

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
