# Peer review of "Morphological and Functional Remodeling of Vascular Endothelium in Cardiovascular Diseases"

_ijms, 2023, doi:10.3390/ijms24031998_

Round 1
Reviewer 1 Report
This is an excellent review about the physiology of endothelium. The authors summarized a lot of details about the vascular endothelium, which helps us to better understand its role in cardiovascular pathology and to determine its involvement in many diseases. It is worth to be published and learned by more readers.
I have no comments on this manuscript.
Author Response
We thank the reviewer for his comments.
Reviewer 2 Report
The authors have provided an adequate review.
Just one minor improvement.
Page 6: please provide reference for DIPLO.
Author Response
We thank the reviewer for his helpful comments. As suggested we added the references of Baker as well as of Dipolo.
Reviewer 3 Report
The paper : ”Morfological and functional remodeling of vascular endothelium in cardiovascular diseases” is an interesting review about the role of the endothelium as an active protagonist of physiological and pathological processes. Although the topic is very important, nevertheless some little things in the paper must be improved. I don't understand the order of the paragraphs: why do paragraphs 3-. Role of the endothelium in vascular physiology 4. Structure of the vascular endothelium of arteries and veins 5. Role of the endothelium in vascular activity….6. precede the more general paragraph 6. The vascular endothelium? Please move up the paragraph 6.
The quality of Figures 1 and 2 is very low, please improve them.
Author Response
We thank the reviewer for his helpful comments. As suggested, we displace section 6 as new section 2. Unfortunately, the figures were saved as 300 dpi, as suggested by the journal, and we can not do better. We thank you again for your help.